# Alterations in the Peritoneal Fluid Proteome of Horses with Colic Attributed to Ischemic and Non-Ischemic Intestinal Disease

**DOI:** 10.3390/ani15111604

**Published:** 2025-05-30

**Authors:** Rebecca C. Bishop, Justine V. Arrington, Pamela A. Wilkins, Annette M. McCoy

**Affiliations:** 1Department of Veterinary Clinical Medicine, University of Illinois, 1008 W Hazelwood Dr, Urbana, IL 61802, USA; pawilkin@illinois.edu (P.A.W.); mccoya@illinois.edu (A.M.M.); 2Proteomics Core Facility, Roy J. Carver Biotechnology Center, University of Illinois, 1206 W Gregory Dr, Urbana, IL 61802, USA; jarrington@salk.edu

**Keywords:** equine, colic, abdominocentesis, fluid proteomics

## Abstract

Colic is one of the most devastating diseases faced by horses and their owners. Horses that require surgical treatment for colic are at risk for development of post-operative complications and death. Unfortunately, there are no objective tests that can reliably predict which horses are most likely to develop complications after surgery. Peritoneal fluid (PF) bathes the abdominal organs, and changes in the PF environment likely play an important role in the intestinal diseases associated with colic. Existing tests based on PF are inconsistent at determining prognosis for horses with colic. This study aimed to quantify differences in PF proteome (the complete set of proteins expressed within the fluid) between horses with colic caused by different intestinal lesions. PF fluid samples were collected from horses presenting to a hospital with signs of colic, and cases were categorized by lesion type (ischemic or non-ischemic) and location (small or large intestine). Identified proteins related to ischemic and small intestinal lesions were associated with immune and inflammatory responses, as well as lipid metabolism pathways not previously described in ischemic intestinal disease. These proteins should be considered as candidates for further study to improve the ability of equine veterinarians to care for horses with colic.

## 1. Introduction

Colic (abdominal pain) is one of the most common causes of equine morbidity and mortality, yet there is still much to be learned about its etiology and pathophysiology. Survival and complication rates of horses with colic vary with the type of lesion and segment of gastrointestinal tract involved [1,2,3,4]. Horses that require surgical treatment, especially when resection and anastomosis is performed for treatment of ischemic intestine, have increased risk of complications and decreased survival when compared to horses with colic lesions that can be managed without surgery [5,6,7,8,9,10]. In some studies, lesions affecting the small intestine (duodenum, jejunum, or ileum) are associated with a worse prognosis than large intestine (cecum, large colon, or small colon) [4,11]. Post-operative ileus [4,12], recurrent colic [13], and repeat laparotomy [14] are all associated with mortality in the early post-operative period. Despite advances in diagnostic modalities, there remains no reliable way to identify which horses treated for colic are at higher risk of complications.

The peritoneal environment likely plays an important role in the onset and/or progression of intestinal disease as well as post-operative complications, yet it is relatively understudied. Peritoneal fluid (PF) is an ultrafiltrate of plasma that lubricates abdominal organs. Due to its intimate association with the gastrointestinal tract and other abdominal organs, PF more directly reflects abdominal pathology than serum or plasma. Measurable factors in PF that have been described as markers of intestinal disease severity include total protein concentration, L-lactate, acute phase proteins, and alkaline phosphatase [15,16,17,18,19]. However, these are crude measures that have had inconsistent success at predicting post-operative complications and survival. It is unlikely that a single marker will be sufficiently sensitive or specific to accurately reflect pathology. Quantitative evaluation of the PF proteome provides a more comprehensive view of the health of the peritoneal cavity compared to more traditional clinical measurements.

High throughput proteomics has been used to characterize the protein composition of various body fluids to better understand disease pathophysiology and to identify potential biomarkers for diagnosis or monitoring of disease in humans and animals [20,21,22,23,24]. An “Equine peptide atlas” of 2636 canonical proteins has been created to facilitate proteomics by high-resolution tandem mass spectrometry (MS) [25]. A single study has assessed the proteome of equine PF [26]. Using label-free MS, 45 proteins were identified as differentially abundant between healthy controls and horses with strangulating small intestinal disease (SSID). These proteins had a range of functions including immune modulation, cell adhesion, and hemostatic balance [26]. Twelve proteins were also differentially abundant between non-surviving and surviving horses with SSID, suggesting a continuum of abundance for some proteins between healthy horses, survivors, and non-survivors. However, this study utilized dynamic range compression to deplete high abundance proteins, limiting the ability to draw conclusions regarding quantitative differences between groups. Further investigation of the PF proteome in abdominal disease is warranted. While whole-proteome sequencing of PF is not practical in a clinical setting, characterizing PF protein profiles associated with different types of intestinal disease is a critical first step towards identifying novel candidate biomarkers that may have future diagnostic or prognostic utility.

The objectives of this study were to quantify differences in PF protein abundance between horses with colic attributed to ischemic or non-ischemic intestinal pathology and between horses with lesions affecting the small intestine (SI) or large intestine (LI). PF samples were collected at admission from 20 adult horses presenting with signs of colic and processed by chloroform–methanol precipitation followed by Lys-C and trypsin digestion to extract peptides. We used a label-free liquid chromatography-tandem mass spectrometry (LC-MS/MS)-based approach to quantify differential protein abundance. We hypothesized that PF proteome signatures would discriminate between both the type of intestinal disease and portion of intestine affected (SI vs. LI).

## 2. Materials and Methods

### 2.1. Sample Collection and Processing

Peritoneal fluid (PF) samples were collected from client-owned horses presented to the University of Illinois Veterinary Teaching Hospital with signs of colic. All study procedures were approved by the University of Illinois IACUC (protocols #19092 and #22030) and informed client consent was obtained at the time of study enrollment. Adult (≥2 years of age) horses with no history of abdominal surgery in the previous 30 days were eligible for inclusion. Horses with a clinical diagnosis of colitis or primary peritonitis were excluded. PF samples were collected from horses undergoing abdominocentesis as part of the clinical evaluation for colic. Horses were restrained with standing sedation at the discretion of the attending clinician, and abdominocentesis was performed following standard clinical protocols [27]. PF was collected into plain sterile blood collection tubes, centrifuged at 1500× *g* for 4 min, and supernatant was aliquoted for storage at −80 °C.

### 2.2. Clinical Data Collection

Patient signalment, history, clinical evaluation parameters, diagnosis, treatment, complications, and short-term survival were recorded for each horse. Clinical evaluation parameters included basic physical examination parameters (heart rate, temperature, respiratory rate, mucous membrane appearance, and capillary refill time); stall-side bloodwork (packed cell volume, total solids, and blood L-lactate); complete blood count, fibrinogen, and serum chemistry (if available); and stall-side peritoneal fluid analysis (gross appearance, total solids, and lactate). Other routine diagnostics for clinical evaluation included transrectal palpation, transabdominal ultrasound, and nasogastric intubation at the discretion of treating clinicians, but were not included for analysis. Treatment was categorized as medical (typically hospitalization with intravenous/enteral fluids and pain management), surgical (exploratory laparotomy, performed due to clinical identification of a surgical lesion and/or persistent pain in the face of analgesia), or euthanasia (if patient was euthanized at the time of admission without attempted medical or surgical treatment). After case resolution (discharge or death/euthanasia), horses were assigned to groups based on lesion type (non-ischemic (NS) or ischemic disease) and lesion location (LI or SI), utilizing the clinical and/or post-mortem diagnosis as recorded in the medical record. Samples were included such that there was an even distribution of lesion locations within each lesion type (*n* = 5 each: NS-LI, NS-SI, ischemic-LI, ischemic-SI). Additional clinical data, including venous blood gas analysis, packed cell volume and total solids, PF L-lactate, PF total solids, and PF gross appearance, were recorded retrospectively from the electronic medical record system if available [28].

### 2.3. Protein Extraction

Frozen PF samples were submitted to the University of Illinois Roy J. Carver Biotechnology Center Proteomics Core Facility for processing and liquid chromatography-tandem mass spectrometry (LC-MS/MS) following standard protocols [29]. The protein content in the peritoneal fluid samples was measured by bicinchoninic acid (BCA) assay (Pierce). The samples were treated with 10 mM tris(2-carboxyethyl)phosphine (TCEP) to reduce the protein disulfide bonds and 40 mM 2-chloroacetamide (CAA) was added to alkylate the reduced cysteine residues. The samples were then heated to 95 °C for 15 min. After cooling, proteins were extracted by chloroform–methanol precipitation, and the protein pellets were briefly dried. The protein pellets were reconstituted in 100 mM triethylammonium bicarbonate (TEAB) and digested by LysC (1:100 enzyme:substrate *w*/*w*) at 28 °C for 3 h. Further digestion was performed with trypsin (1:50 *w*/*w*) overnight at 37 °C. On the following day, the digestion was stopped with 10% trifluoroacetic acid (TFA), desalted with StageTips [30], and dried. Prior to LC-MS analysis, the peptide amount was again normalized according to a BCA assay.

### 2.4. Liquid Chromatography-Mass Spectroscopy

The dried peptides were suspended in 5% acetonitrile (ACN) with 0.1% formic acid (FA), and 1 µg of peptides from each sample was injected into an UltiMate 3000 RSLC nanoflow system coupled to a Q Exactive HF-X mass spectrometer (Thermo Scientific, Waltham, MA, USA). Peptides were separated in the reversed phase using a 25 cm Acclaim PepMap 100 C18 column (2 µm particle size, 75 µm ID) maintained at 50 °C and mobile phases of 0.1% FA (A) and 0.1% FA in 80% ACN (B). The gradient ran from 5% B to 35% B over the course of 60 min, followed by an increase to 50% B over 10 min. The column was then washed and equilibrated prior to the next run. The mass spectrometer was operated in the positive polarity, and full MS scans from 350 to 1500 *m/z* were collected at 120 k resolution. The top 15 ions from each MS scan were subjected to higher energy collisional dissociation (HCD) at 30 nominal collision energy. MS2 scans of the resulting fragment ions were collected at 15 k resolution. The isolation window was set to 1.2 *m/z*, and the dynamic exclusion time was set to 60 s.

### 2.5. Protein Identification and Functional Annotation

The raw LC-MS data was processed with MaxQuant v2.010 to both identify and quantify the proteins. Settings for the MaxQuant searches included peptide mass tolerances of 20 ppm and 4.5 ppm for the first and main searches, respectively. The fragment mass tolerance was set to 20 ppm. A tryptic digest with a maximum of 2 missed cleavages was specified along with a fixed modification for cysteine carbamidomethylation and variable modifications of protein N-terminal acetylation, methionine oxidation, and asparagine/glutamine deamidation. Searches were performed against the Uniprot *Equus caballus*, *Bos taurus*, and common lab contaminants databases. The false discovery rate (FDR) at the peptide spectrum matches and protein levels was set to 1%, using the fast LFQ algorithm with a minimum of 2 peptide ratios and the match between runs feature enabled with a 1 min time window.

A total of 262 proteins were identified, which was reduced to 236 after removal of contaminants. The proteomics data, including raw files, MaxQuant parameters, and results, have been deposited in the ProteomeXchange Consortium via the jPOST partner repository [31] with the data set identifiers JPST003601 and PXD060655.

### 2.6. Statistical Analysis

Further data processing was performed in Excel (Microsoft Corporation, Redmond, WA, USA) and R version 4.3.1 [32] using RStudio version 2022.12.0 (PBC, Boston, MA, USA). Data manipulation was performed with ‘dplyr’ version 1.1.4 [33] and ‘magicfor’ version 0.1.0 [34]. Continuous clinical data were summarized as median (quartile 1, quartile 3), and normality was assessed by Shapiro–Wilk test and visual inspection. The effect of lesion type and location on continuous clinical data was assessed by two-way ANOVA with post-hoc Tukey’s HSD as appropriate using ‘rstatix’ version 0.7.2 [35]. Effect size was described by eta^2^ and a *p*-value of <0.05 was considered significant. Categorical data were summarized as count data.

Qualitative analysis of proteomic data was completed using Venny version 2.1 [36] to identify proteins unique to each group within lesion type (Ischemic vs. NS) and lesion location (LI vs. SI). Unique proteins within each group were entered in STRING version 11.5 (https://version-11-5.string-db.org/) to visualize protein–protein interactions and assess functional enrichments, with a minimum required interaction score of 0.4. Functional enrichments (strength ≥ 1) were identified from Gene Ontology (Molecular Function and Cellular Component), local network cluster (STRING), subcellular localization (COMPARTMENTS), KEGG pathways, and annotated keywords (UniProt). For all analyses, FDR < 0.05 was considered significant.

For quantitative analysis, removal of proteisns with an LFQ intensity of zero further reduced the protein set to 151. Differential abundance analysis was performed with ‘DESeq2’ version 1.40.2 [37] and results were visualized using the ‘EnhancedVolcano’ package version 1.18.0 [38]. Given the small sample size, proteins were considered “of interest” based on a raw *p*-value ≤ 0.05 and log_2_FC > 0.5 or < −0.5. PCA analysis was performed using the ‘MSnSet.utils’ package version 0.2.0 [39]. Network analysis and functional enrichment of proteins with log_2_FC over the pre-determined threshold for ischemic lesions and SI lesions was completed using STRING, as described for unique proteins above.

To analyze the relationship between clinical data, protein intensities, and lesion type or location, we employed a random forest (RF) model. Missing datapoints were imputed to the population mean using ‘missMethods’ package version 0.4.0 [40]. Random forest analysis was performed with the ‘RandomForest’ package version 4.7-1.2 [41], with ntree = 1000 and node size = 5. The Boruta feature selection algorithm (‘Boruta’ package version 8.0.0 [42]) was used to identify the most influential features. “Confirmed” and “tentative” variables were included in reduced RF models for prediction of lesion type and lesion location. Variables with greater Gini importance were considered to contribute more significantly to the model’s predictive ability. Out-of-bag (OOB) error was used to assess the ability of the model to discriminate between groups, with OOB error below 30% considered good [43].

## 3. Results and Discussion

### 3.1. Clinical Data

The study population included six Quarter Horses, two each Arabian, Missouri Fox trotter, Morgan, Standardbred, and Thoroughbred, and one each Belgian, Clydesdale, Mustang, and Pony of the Americas. There were 9 geldings and 11 mares. Clinical data are summarized in Table 1. While there was not a statistically significant effect of group on age, horses with SI lesions were older than those with LI lesions, correlating with the increased incidence of small intestinal strangulating lipomas in aged horses (typically > 15 years of age) [2,44]. Capillary refill time was greater in horses with ischemic SI lesions (*p* = 0.005), which likely reflects increased incidence of systemic inflammatory response syndrome (SIRS) in these patients. Venous L-lactate, PF L-lactate, and PF total solids were also significantly different between groups, supporting the well-established association between hyperlactatemia and ischemic intestinal injury [45,46,47]. PF was described as serosanguinous in 5/5 horses with ischemic SI lesions, and 3/5 with ischemic LI lesions; turbid in 1 horse with ischemic LI; slightly turbid in 2/5 horses with NS SI lesions; and within normal limits in 3/5 NS SI and 5/5 NS LI lesions. Clinically, PF with a serosanguinous appearance is strongly associated with ischemic intestinal lesions [15,48,49].

By study design, horses were evenly distributed between groups by lesion type and location. Specific diagnoses, treatment, and outcomes are summarized in Table 2. The overall survival rate was 10/20 horses (50%), with most non-survivors having ischemic lesions (5/5 ischemic SI, 4/5 ischemic LI). While the correlation between changes in PF proteins and outcomes is of great clinical interest, the degree of collinearity between survival and lesion type prevented meaningful analysis of survival as an outcome in this study. The majority of non-survivors (including all horses with ischemic SI lesions) were euthanized based on the decision of the owner not to proceed with a recommended exploratory surgery, or not to continue after confirmation of ischemic intestine at the time of surgery. Unfortunately, the rate of financially motivated euthanasia in this study is reflective of the general colic population at our hospital, and it is impossible to determine which of these horses may have survived with continued treatment.

### 3.2. Unique Proteins

Of the 236 unique proteins identified, 206 were found in all groups (Figure 1). Looking at groups based on lesion type (N = 10 each), there were 16 proteins found only in horses with ischemic lesions and 11 found only in horses with NS lesions. When horses were grouped by lesion location, there were 8 proteins found only in horses with LI lesions and 12 proteins unique to SI lesions. The identities of unique proteins are summarized in Appendix A.

The unique proteins associated with ischemic and SI lesions had significant protein–protein interaction enrichment (*p* < 0.001 for both), while NS lesions (*p* = 0.36) and LI lesions (*p* = 0.2) did not (Table 3, Figure 2).

Notable functional enrichments within ischemic lesions included multiple terms related to the inflammatory cascade: neutrophil and complement activation (CL:15549), calprotectin complex (GOCC:1990660), TLR_4_/NF-κB signaling (S100A8 and S100A9 complexes, GOCC:1990661 and GOCC:1990662), and KEGG pathway for IL-17 signaling (ecb04657). Among other functions, TLR_4_/NF-κB signaling is responsible for endotoxin-mediated activation of innate immune responses. Horses are both exquisitely sensitive to the effect of endotoxin in the systemic circulation and at risk of endotoxemia with ischemic intestinal lesions due to the vast quantities of Gram-negative bacteria within their intestinal flora [50,51,52,53]. Top GO terms were related to calcium ion binding (molecular function, GO:0005509) and the extracellular region (cellular component, GO:0005576 and GO:0005615).

There was also significant enrichment for the KEGG pathway related to *Salmonella* infection (ecb05132). While the clinical consequence of *Salmonella* infection-related responses within the PF is unknown, this finding is not surprising. *Salmonella* species are found in the intestinal microbiome of some horses, and both clinical manifestations of salmonellosis and increased rates of asymptomatic *Salmonella* shedding are well documented in horses with colic [54,55]. *Salmonella* shedding has also been associated with increased incidence of specific complications (surgical site, weight loss) in horses recovering from colic surgery [56].

Proteins with functions related to inflammation and immune response are of particular interest, as there is a clear connection to physiological functions that are likely occurring in the equine acute abdomen: responding to or protecting against bacterial translocation through damaged intestinal walls [57,58], reflecting ongoing inflammation [59], or identifying pathogen-associated molecular patterns to further increase inflammatory responses [60]. Annexin (ANXA2), lactotransferrin (LTF), and leukocyte elastase inhibitor (SERPINB1) have direct or indirect antimicrobial activity. LTF has been used as a biomarker for gastrointestinal inflammation in other species [61,62] and it may be related to immune modulation to maintain gut homeostasis [63,64]. Other proteins, such as pentaxin (PTX3) and peptidoglycan-recognition protein (ENSECAP00000015762), act as pattern recognition receptors to alert the immune system to the presence of bacteria. Peptidase S1 family proteins (CRTC, ENSECAP00000012520) participate in immunoglobulin-A-mediated immune responses.

Galectin-3 binding protein (LGALS3BP) is also of interest as it has multiple roles including stimulation of IL-6 expression in stromal cells, and regulation of cell–cell and cell–matrix adhesion [65,66]. In human intestinal epithelial cells, reduced expression of LGALS3BP has been associated with reduced intercellular adhesion [67]. However, galectins more generally can act as both agonists and antagonists of cellular adhesion, depending on the context (level of expression, location, receptor type, and glycosylation) [68,69]. In the previous proteomic study of equine peritoneal fluid, galectin-3 binding protein was differentially abundant between healthy controls and horses with strangulating small intestinal disease, as well as between survivors and non-survivors, with the lowest expression in non-survivors with SSID (compared to survivors and healthy controls) [26].

In the group of proteins unique to SI lesions (compared to LI), enriched functions largely overlapped with those identified in ischemic lesions. This result is most likely explained by the fact that 8 of the 12 proteins in this group were specifically unique to ischemic SI lesions (and thus, were included in both analyses). Enriched functions were primarily related to inflammation, including neutrophil aggregation and complement activation (CL:15549, CL15546), calprotectin complex (GOCC:1990660), S100A8 (GOCC:1990661), and S100A9 complexes (GOCC:1990662), as seen in the ischemic network. Calprotectin (S100A8/S100A9 complex, a member of the S100 protein family), is a cytoplasmic protein primarily found in neutrophils, but also within circulating monocytes and keratinocytes, among others, and is typically released after cell disruption and death. Calprotectin is of interest in equine intestinal disease because of its use as a biomarker of inflammatory bowel disease in humans. In equine studies, increases in calprotectin abundance have been associated with experimental large colon ischemia (colonic venous blood) [70], clinical and experimental SI ischemia (full thickness biopsy with immunohistochemistry) [71], colitis (feces) [72], and equine gastric ulcer syndrome (saliva) [73]. Increased concentration of calprotectin in ascitic fluid is associated with cirrhosis- and immunodeficiency-associated spontaneous bacterial peritonitis in humans [74,75]. No studies reporting calprotectin concentrations in equine PF have been published to date.

### 3.3. Differential Protein Abundance

There were 129 proteins with sufficient abundance for inclusion in differential abundance analysis. Based on a cutoff of FDR < 0.05, there were no proteins with significant differential abundance between lesion type or location (Appendix A). As the objective of this study was to identify proteins of interest for further study in a larger population, proteins were considered “of interest” based on *p*-value < 0.05 and log_2_FC > 0.5 or <−0.5 (Figure 3).

Two proteins had increased abundance in horses with NS lesions, Fetuin B (FETUB) and a peptidase SI family protein (CFD). FETUB and related proteins have been implicated in range of functions, including but not limited to insulin regulation and response to systemic inflammation [76,77]. In pediatric patients with acute abdominal pain, decreased serum concentrations of fetuin A were associated with acute appendicitis and perforation, compared to other abdominal diseases [78]. Increases in serum FETUB have been associated with intense exercise training [79] and endometritis in horses [80], but to our knowledge, fetuins have not been evaluated in horses with colic. Possible physiologic roles of peptidases are discussed below (see ‘Random Forest Classification’).

In horses with ischemic lesions, proteins with increased abundance included alpha-2-macroglobulin (A2M), immunoglobulin-like domain-containing protein (ENSECAP00000012711), apolipoprotein B (APOB), and two uncharacterized proteins (ENSECAP00000007504 and ENSECAP00000029691). A2M is a protease inhibitor and is discussed further under ‘Random Forest Classification’ below. APOB is typically considered for its role in chylomicron formation and lipid transport. APOB also has high-affinity binding sites for heparin, and heparin binding results in activation of lipoprotein lipase and increases in triglyceride and lipoprotein clearance [81]. The effect of APOB binding on heparin’s anticoagulant properties is unknown. Another apolipoprotein, APOA-IV, has known anti-inflammatory properties and upregulation has been described in horses with chronic laminitis [82].

Horses with SI lesions also had increased abundance of APOB, along with apolipoprotein C2 (APOC2) and hemoglobin subunits alpha (HBA2) and beta (HBB), and one of the same uncharacterized proteins as in horses with ischemic lesions (ENSECAP00000029691). The increased abundance of HBA2 and HBB in horses with SI lesions is unsurprising when considering that the differential abundance changes in this group seem to be driven by the ischemic-SI lesion subset. Clinically, bloody or serosanguinous peritoneal fluid is known to be associated with strangulating lesions, and among cases in this study, the ischemic-SI cases had more serosanguinous PF than ischemic-LI cases. Interestingly, HBB, LTF, and A2M were each found to be downregulated in the saliva of horses with colic in a previous study, compared to healthy controls [83].

Principal component analysis demonstrated a distinct cluster of horses with ischemic SI lesions, but there was significant overlap between NS SI lesions and ischemic LI lesions (Figure 4). Subjectively, the extent of ischemic intestinal injury in SI lesions at the time of presentation is often greater than the overall severity of LI ischemic injury. NS LI lesions formed a tight cluster but also had some overlap with ischemic LI and NS SI lesions. It is possible that evaluating groups based on more specific diagnoses would demonstrate improved segregation, but a larger sample size would be required.

### 3.4. Functional Enrichment

Functional enrichment analysis was performed in STRING for proteins associated with ischemic lesions and SI lesions based on differential abundance analysis. There were 61 proteins included in the network for ischemic lesions, which had 438 predicted edges, average local clustering coefficient of 0.6, and showed significant functional enrichment (*p* < 1 × 10^−16^), with 121 functional enrichments identified. There were 59 proteins included in the network for SI lesions, which had 418 edges, average local clustering coefficient of 0.62, and significant functional enrichment (*p* < 1 × 10^−16^), with 123 functional enrichments identified.

There was significant functional overlap for ischemic and SI networks, with 103 functional enrichments identified in both networks (Appendix A). This likely reflects that ischemic SI lesions specifically were driving the differential abundance in both analyses, as demonstrated by PCA. The primary themes of shared functions revolved around lipoprotein metabolism/clearance and immune responses (complement activation, immunoglobulins). Upregulation of immune and inflammatory responses is unsurprising given the profound systemic response to the presence of ischemic bowel, and in some cases of NS SI disease (i.e., enteritis). To our knowledge, little is known about the role of lipoproteins in peritoneal fluid or in relation to intestinal disease, highlighting an interesting area for future study.

There were 20 functional enrichments unique to ischemic lesions (Table 4). Based on enrichment strength, the top GO terms for biological process were cytolysis (GO:0019835), regulation of lipoprotein lipase activity (GO:0051004), triglyceride homeostasis (GO:0070328), and negative regulation of blood coagulation (GO:0030195). The top GO terms for molecular function were lipoprotein lipase activator activity (GO:0060230), complement binding (GO:0001848), and heparan sulfate proteoglycan binding (GO:0043395). Local network cluster enrichment was related mixed pathways including high-density lipoprotein particle receptor binding, and blood coagulation, intrinsic pathway (CL:17303), regulation of exo-alpha-sialidase activity and positive regulation of extracellular matrix constituent secretion (CL:17333), and complement activation, lectin pathway, and synapse pruning (CL:17492). There was also enrichment in the KEGG pathway for phagosome (ecb04145).

There were 18 functional enrichments unique to SI lesions (Table 5). The top GO terms for biological process were low-density lipoprotein particle clearance (GO:0034383), positive regulation of lipid storage (GO:0010884), and acute-phase response (GO:0006953). There were multiple enriched biological process and function terms related to peptidase/protease regulation: negative regulation of peptidase activity (GO:0010466, GO:0010951), and protease binding (GO:0002020). Local network cluster, cellular component, and subcellular localization enrichments were related to coagulation: fibrinogen complex (GO:0005577, GOCC:0005577), complement and coagulation cascades and protein–lipid complex (CL:17287), and platelet alpha granule (GOCC:0031091).

### 3.5. Random Forest Classification

A random forest classification model for classification by lesion type, generated from a data set including all expressed proteins and clinical data had an overall OOB error rate of 35%, with class error rate of 40% for ischemic lesions and 30% for NS. Based on Boruta feature selection, five features were classified as “confirmed important” and nine as “tentative” (Table 6). The random forest model consisting of only these 14 features had improved OOB error rate at 25%, attributed to improvement in classification of ischemic lesions (20% class error vs. 30% for NS lesions).

Clinical variables included in the final model for classification of lesion type were PF color, PF total solids, and venous L-lactate. This supports the clinical practice of using PF color as a gross indicator of intestinal ischemia. The inclusion of PF total solids and venous L-lactate is also unsurprising, as both were significantly different in horses with ischemic lesions in univariate analysis and are consistently associated with ischemic lesions in other studies [15,48,84,85,86].

Several of the proteins in the final model were related to blood and blood clotting, including HBA2 and HBB (also identified in DE analysis), albumin (ALB), plasmin (PLG), and A2M. As direct blood contamination was removed during pre-analytical sample processing, the presence of hemoglobin subunits suggests extravasation of blood components and lysis of red blood cells. Plasmin plays a key role in fibrin clot dissolution and has a positive feedback effect on plasminogen activators. Plasmin-mediated cleavage of other proteins, such as fibronectin and laminin, results in cell detachment, apoptosis, and inflammation [87].

There were also multiple proteins with serine inhibitor functions, including A2M, alpha-1-antiproteinase 2 (SPI2), and inter-alpha-trypsin inhibitor heavy chain 4 (ITIH4). Serine proteases are a large family of enzymes with roles in blood clot formation, apoptosis, inflammation, and digestion [88,89]. As regulation of these proteases by their inhibitors is essential to prevent damage from excessive proteolysis [89], increased abundance of serine protease inhibitors in horses with ischemic lesions may reflect an appropriate response to intestinal damage. However, dysregulation of protease activity is also associated with fibrosis, and could contribute to adhesion formation. Serine protease inhibitors have been used pharmacologically in humans to prevent or delay fibrosis associated with diseases such as chronic renal failure [90], atrial fibrillation [91], and idiopathic pulmonary fibrosis [92].

The group of proteins included in the lesion type classification model had significant functional enrichment (*p* < 1 × 10^−8^) with an average local clustering coefficient of 0.49. The top GO terms for molecular function were oxygen carrier activity (GO:0005344), oxygen binding (GO:0019825), and serine-type endopeptidase inhibitor activity (GO:0004867). There was enrichment in both the local network cluster and KEGG pathways for complement and coagulation cascades (CL:17287, CL:17289, ecb04610).

A random forest classification model for classification by lesion location, generated from a data set including all expressed proteins and clinical data had an overall OOB error rate of 35%, with class error rate of 40% for LI lesions and 30% for SI. Based on Boruta feature selection, four features were classified as “confirmed important” and three as “tentative” (Table 7). The random forest model consisting of only these seven features had improved OOB error rate at 15%, attributed to improvement in classification of both LI (20% class error) and SI lesions (10%). However, clinical differentiation between SI and LI lesions is relatively straightforward, and classification by protein features is unlikely to augment existing clinical practice.

Age was the only clinical variable retained in the random forest model likely attributed in part to the greater age of horses with SI lesions compared to LI. Included proteins were associated with blood components and coagulation, including hemoglobin subunit theta-1 (HBA), fibrinogen alpha and beta chains (FGA, FGB), and ALB. As seen in differential abundance analysis, APOB was also included in the final classification model.

The group of proteins included in the lesion location classification model had significant functional enrichment (*p* < 1 × 10^−7^) with an average local clustering coefficient of 0.75. There was local network cluster enrichment related to lipoprotein binding and regulation (CL:17300, CL:17292) and complement and coagulation cascades, also associated with lipoprotein particles (CL:17289). As for the lesion type model, there was enrichment in the KEGG pathway for complement and coagulation cascades (ecb04610).

### 3.6. Limitations

Biological variability is a major limitation of this study, as classification of colic lesions into “ischemic” or “non-ischemic” likely does not capture differences that would be identified between more granular categories. However, the modest sample size limits the ability to make meaningful comparison between more narrowly defined groups (such as ischemic lesion survivors vs. non-survivors). The small number of surviving horses and uneven distribution of survivors among groups also prevented investigation of protein profiles associated with outcome or complications Additionally, a label-free LC-MS-based approach without dynamic range compression was selected to provide a biologically accurate view of protein abundance. However, the presence of high concentration blood proteins may have limited the detection range and excluded some low-abundance proteins.

## 4. Conclusions

Overall, differential abundance, network analysis, and random forest classification identified consistent themes of protein abundance and functional enrichment in the peritoneal fluid of horses with colic attributed to ischemic and SI lesions, in comparison to non-ischemic and LI lesions. Hemoglobin subunits alpha and beta were associated with SI and ischemic lesions, consistent with known clinical correlation between serosanguinous PF and ischemic intestine. Several of the identified proteins and pathways were associated with well-described immune and inflammatory responses, many not previously assessed in equine PF directly, such as calprotectin, lactotransferrin, alpha-2-macroglobulin, and the complement and coagulation cascades. Apolipoprotein B and lipid metabolism pathways were also associated with ischemic lesions in multiple analysis, which has not previously been described. While no single biomarker is expected to adequately diagnose or predict the outcome of equine colic, the features identified through this study should be considered as candidates for further study in a larger population.

## Figures and Tables

**Figure 1 animals-15-01604-f001:**
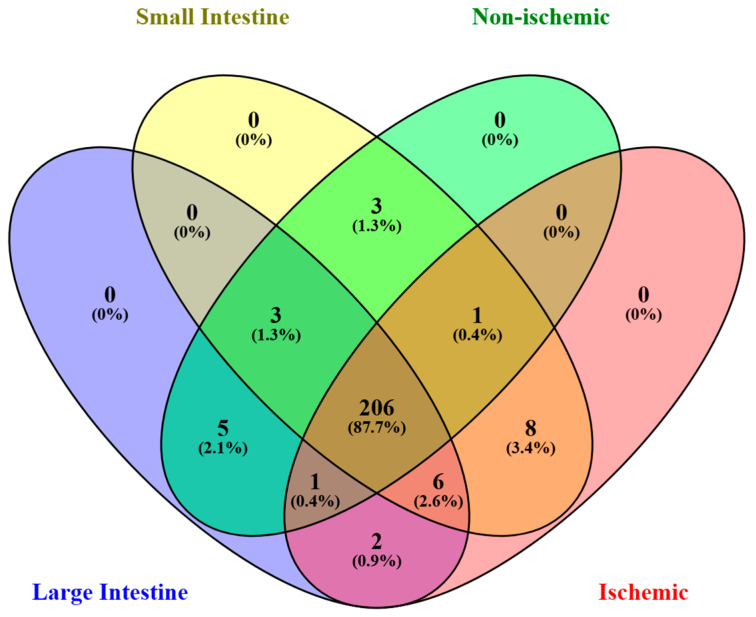
Venn diagram showing the overlap between proteins identified in peritoneal fluid of horses with colic attributed to ischemic or non-ischemic lesions of the small intestine or large intestine.

**Figure 2 animals-15-01604-f002:**
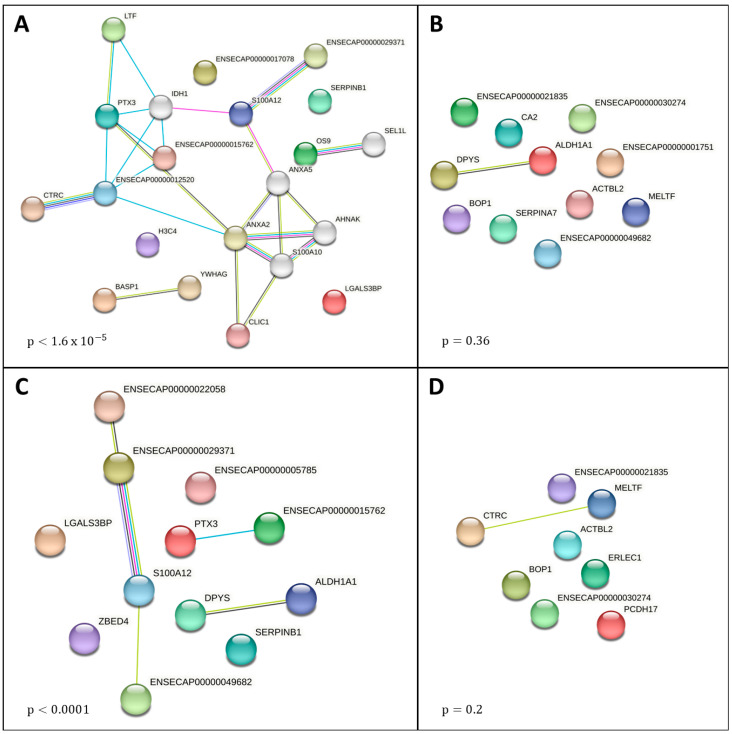
Functional network diagrams from network analysis by STRING v11.5 for unique proteins found in the peritoneal fluid of horses with (**A**) ischemic intestinal lesions, (**B**) non-ischemic intestinal lesions, (**C**) small intestine lesions, and (**D**) large intestine lesions. Each network node represents proteins, labeled by the protein coding gene locus, and edges represent known and predicted protein–protein interactions. The *p*-value reflects protein–protein interaction enrichment, with *p* < 0.001 meaning that the network has significantly more interactions than would be expected for a random set of proteins of the same size and degree distribution, suggesting that the proteins as a group are biologically connected.

**Figure 3 animals-15-01604-f003:**
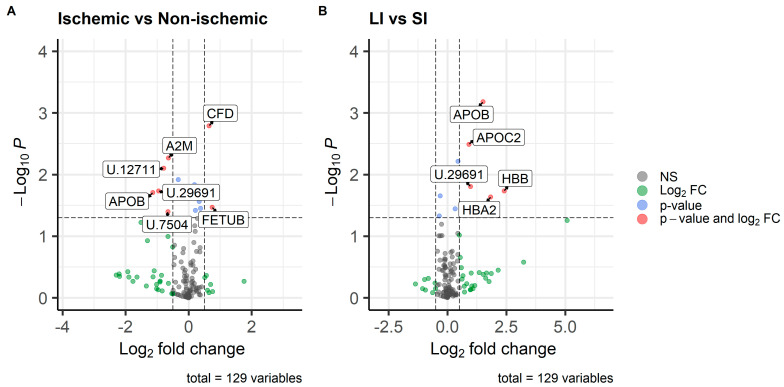
Volcano plots demonstrating relative abundance of proteins in peritoneal fluid of horses with colic, generated using a two-sided *t* test: (**A**) lesion type, (**B**) lesion location. Dashed lines represent cutoffs based on *p*-value < 0.05 and Log_2_ fold change < −0.05 or >0.05. Proteins considered of interest are colored in red and labeled with the gene name (U. = ENSECAP000000 for uncharacterized proteins).

**Figure 4 animals-15-01604-f004:**
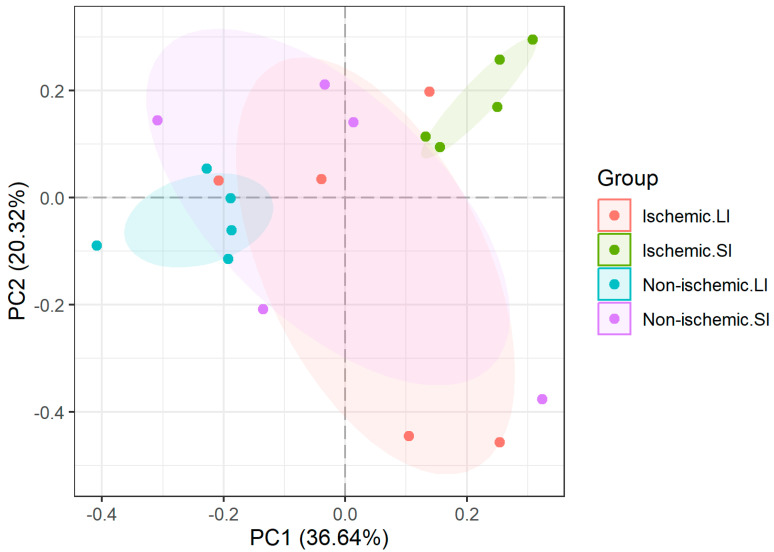
Principal component analysis (PCA) of protein abundance in peritoneal fluid from horses with colic, grouped by lesion type (ischemic vs. non-ischemic) and location (large intestine (LI) vs. small intestine (SI)). PCA was based on the data set containing log_2_ transformed normalized reported ion intensities, filtered for proteins with missing abundance.

**Table 1 animals-15-01604-t001:** Summary of continuous clinical data from horses evaluated for colic. Data are summarized as median (quartile 1, quartile 3). The result of one-way ANOVA for effect of group is summarized by effect size (eta^2^) and *p*-value. Superscript letters indicate significant differences between groups based on post-hoc Tukey’s HSD; groups sharing a letter were not significantly different.

Variable	N	All Horses	Ischemic		Non-Ischemic		ANOVA	
			LI	SI	LI	SI	eta^2^	*p*-Value
Age (years)	20	14 (7.8, 20)	8 (6, 9)	20 (17, 22)	7 (4, 9)	18 (17, 20)	0.36	0.06
Weight (kg)	17	500 (454, 542)	500 (500, 516)	403 (315, 429)	500 (485, 542)	517 (481, 561)	0.44	0.05
Heart rate (bpm)	20	60 (44, 80)	56 (48, 60)	80 (80, 84)	56 (44, 60)	44 (38, 80)	0.32	0.09
Respiratory rate (brpm)	18	24 (16, 39)	17 (16, 23.5)	50 (36, 60)	20 (16, 36)	24 (16, 30)	0.41	0.06
Temperature (°F)	16	100.3 (99.1, 101)	99.8 (98.7, 100.4)	101.2 (100.2, 101.3)	100.5 (100.2, 100.6)	99.4 (98.1, 100.1)	0.10	0.7
CRT (sec)	20	2 (1, 3)	2 (2, 3) ^a,b^	3 (3, 4) ^a^	1 (1, 1) ^b^	2 (1, 2.5) ^a,b^	0.47	0.02 *
PCV (%)	20	41 (34, 51)	36 (32, 43)	53 (42, 55)	34 (34, 40)	44 (39, 47)	0.21	0.3
TS (g/dL)	16	7.4 (6.4, 8)	6.8 (5.9, 7.4)	8.0 (7.0, 8.4)	7.8 (7.6, 8.0)	6.6 (5.8, 7.0)	0.0	0.4
Lactate (mmol/L)	17	2.2 (1.4, 2.9)	2 (1.4, 3.8) ^a,b^	7.7 (2.9, 7.8) ^a^	1.6 (0.9, 1.9) ^b^	2.4 (2, 2.6) ^a,b^	0.46	0.04 *
PF lactate (mmol/L)	20	2.0 (1.3, 6.3)	1.9 (1.3, 6.3) ^a^	11.5 (6.3, 13.2) ^b^	1.9 (1.3, 1.9) ^a^	1.9 (1, 3.3) ^a^	0.66	<0.001 *
PF TS (g/dL)	18	2 (1.4, 3.5)	2.4 (1.8, 3.3) ^a,b^	4.4 (3.9, 5.1) ^b^	2 (1.8, 2) ^a^	1.4 (1, 1.5) ^b^	0.59	0.005 *

* indicates *p* < 0.05. Abbreviations: bpm, beats per minute; brpm, breaths per minute; CRT, capillary refill time; LI, large intestine; N, number of individuals; PF, peritoneal fluid; SI, small intestine.

**Table 2 animals-15-01604-t002:** Summary of clinical diagnoses, treatment, and short-term outcome for 20 horses evaluated for colic, including collection of peritoneal fluid. Cases were assigned to groups based on lesion location (large intestine, LI; small intestine, SI) and type (ischemic or non-ischemic) after resolution. Treatment was categorized as medical (medical treatment only), surgical (exploratory laparotomy), or euthanasia (elected by owner in lieu of treatment). Survival (yes/no) was defined as survival to hospital discharge. The reason for euthanasia, if applicable, was determined from the medical record.

Lesion Type and Location	Horse ID	Diagnosis	Treatment	Survival	Reason for Euthanasia
Ischemic—LI	H20	Colonic ischemia and rupture secondary to sand impaction	Medical	NO	Elected by owner
	H8	Large colon volvulus	Surgical	YES	
	H9		Surgical	NO	Recurrent colic
	H11		Euthanasia	NO	Elected by owner
	H3	Small colon strangulation—pedunculated lipoma	Surgical	NO	Non-resectable lesion
Ischemic—SI	H6	SI strangulation—gastrosplenic ligament entrapment	Surgical	NO	Elected by owner
	H2	SI strangulation—pedunculated lipoma	Euthanasia	NO	Elected by owner
	H13		Euthanasia	NO	Elected by owner
	H18		Euthanasia	NO	Elected by owner
	H15	SI strangulation—epiploic foramen entrapment	Euthanasia	NO	Elected by owner
Non-ischemic—LI	H4	Large colon impaction	Medical	YES	
	H7		Medical	YES	
	H10		Medical	YES	
	H12	Right dorsal displacement of the large colon	Medical	YES	
	H17		Medical	YES	
Non-ischemic—SI	H5	Ileal impaction	Surgical	YES	
	H14	Enteritis	Medical	YES	
	H16		Medical	YES	
	H1	Functional ileus	Medical	YES	
	H19		Medical	NO	Recurrent colic

**Table 3 animals-15-01604-t003:** Functional enrichment analysis results from STRING v.11.5 found in networks based on proteins found in the peritoneal fluid (PF) of horses with ischemic but not non-ischemic lesions (lesion type: ischemic), and proteins found in the PF of horses with lesions of the small intestine, but not large intestine (lesion location: SI). Functions enriched only in the ischemic lesion unique protein group are highlighted in yellow. Inclusion criteria for enriched functions was FDR < 0.05 and enrichment strength ≥ 1. Gene count reflects the number of proteins in the network (Obs) that are annotated with a given term, compared to the number in the background (Bgnd).

Term ID	Description	Gene Count		Strength	FDR
		Obs	Bgnd		
**Molecular function (Gene Ontology)**					
GO:0005509	Calcium ion binding	7	636	0.95	0.029
**Cellular component (Gene Ontology)**					
GO:0005615	Extracellular space	11	2346	0.88	<0.001
GO:0005576	Extracellular region	8	1384	0.79	0.002
**Local Network Cluster (STRING)**					
CL:17762	Mixed, incl. tertiary granule lumen, and growth of symbiont in host	3	21	2.64	0.011
CL:17770	Mixed, incl. methotrexate binding, and growth of symbiont in host	2	6	2.64	0.013
CL:15549	Neutrophil aggregation, and complement activation, lectin pathway	2	7	2.27	0.003
CL:17616	Mixed, incl. lysosome, and collagen catabolic process	4	165	1.5	0.006
**KEGG Pathways**					
ecb04657	IL-17 signaling pathway	5	274	1.39	0.047
ecb05132	Salmonella infection	3	99	1.17	0.007
**Subcellular localization (COMPARTMENTS)**					
GOCC:1990660	Calprotectin complex	9	1549	2.94	0.005
GOCC:1990661	S100A8 complex	5	507	2.64	0.009
GOCC:1990662	S100A9 complex	2	3	2.64	0.009
GOCC:0005615	Extracellular space	2	6	1.11	0.011
GOCC:0005576	Extracellular region	2	6	0.88	<0.001

**Table 4 animals-15-01604-t004:** Functional enrichment analysis results from STRING v.11.5 found in the network based on 61 proteins upregulated in the peritoneal fluid (PF) of horses with ischemic intestinal lesions, but not in a network based on proteins upregulated in PF of horses with small intestinal lesions. Inclusion criteria for enriched functions was FDR < 0.05 and enrichment strength ≥ 1. Gene count reflects the number of proteins in the network (Obs) that are annotated with a given term, compared to the number in the background (Bgnd).

Term ID	Description	Gene Count		Strength	FDR
		Obs	Bgnd		
**Biological Process (Gene Ontology)**					
GO:0045087	Innate immune response	18	612	1	<0.001
GO:0019835	Cytolysis	4	20	1.84	<0.001
GO:0051004	Regulation of lipoprotein lipase activity	3	15	1.84	0.003
GO:0060191	Regulation of lipase activity	4	53	1.41	0.004
GO:0042632	Cholesterol homeostasis	4	70	1.29	0.01
GO:0070328	Triglyceride homeostasis	3	27	1.58	0.013
GO:0030195	Negative regulation of blood coagulation	3	34	1.48	0.022
**Molecular function (Gene Ontology)**					
GO:0005539	Glycosaminoglycan binding	8	218	1.1	<0.001
GO:0001848	Complement binding	3	17	1.78	0.004
GO:0043395	Heparan sulfate proteoglycan binding	3	20	1.71	0.006
GO:0060230	Lipoprotein lipase activator activity	2	3	2.36	0.012
GO:0004252	Serine-type endopeptidase activity	5	171	1	0.021
**Local network clusters (STRING)**					
CL:17492	Mixed, incl. complement activation, lectin pathway, and synapse pruning	4	15	1.96	<0.001
CL:17303	High-density lipoprotein particle receptor binding, and blood coagulation, intrinsic pathway	3	5	2.31	<0.001
CL:17333	Mixed, incl. regulation of exo-alpha-sialidase activity, and positive regulation of extracellular matrix constituent secretion	2	5	2.14	0.022
**KEGG**					
ecb04145	Phagosome	8	258	1.03	<0.001
**Subcellular localization (COMPARTMENTS)**					
GOCC:0034365	Discoidal high-density lipoprotein particle	2	5	2.14	0.014
**UniProt Keywords**					
KW-0186	Copper	2	14	1.69	0.048

**Table 5 animals-15-01604-t005:** Functional enrichment analysis results from STRING v.11.5 found in the network based on 59 proteins upregulated in the peritoneal fluid (PF) of horses with small intestinal lesions, but not in a network based on proteins upregulated in PF of horses with ischemic intestinal lesions. Inclusion criteria for enriched functions was FDR < 0.05 and enrichment strength ≥ 1. Gene count reflects the number of proteins in the network (Obs) that are annotated with a given term, compared to the number in the background (Bgnd).

Term ID	Description	Gene Count		Strength	FDR
		Obs	Bgnd		
**Biological Process (Gene Ontology)**					
GO:0010466	Negative regulation of peptidase activity	12	203	1.32	<0.001
GO:0010951	Negative regulation of endopeptidase activity	11	196	1.3	<0.001
GO:0015850	Organic hydroxy compound transport	6	115	1.27	<0.001
GO:0006953	Acute-phase response	4	33	1.63	<0.001
GO:0046890	Regulation of lipid biosynthetic process	5	123	1.16	0.005
GO:1905954	Positive regulation of lipid localization	4	68	1.32	0.008
GO:0010884	Positive regulation of lipid storage	3	23	1.66	0.009
GO:0010828	Positive regulation of glucose transmembrane transport	3	35	1.48	0.023
GO:0051180	Vitamin transport	3	37	1.46	0.026
GO:0034383	Low-density lipoprotein particle clearance	2	8	1.95	0.042
GO:0006911	Phagocytosis, engulfment	3	50	1.33	0.05
**Molecular function (Gene Ontology)**					
GO:0002020	Protease binding	5	92	1.28	0.002
GO:0031715	C5L2 anaphylatoxin chemotactic receptor binding	2	3	2.37	0.013
**Cellular Component (Gene Ontology)**					
GO:0005577	Fibrinogen complex	2	2	2.55	0.004
**Local network clusters (STRING)**					
CL:17287	Complement and coagulation cascades and protein–lipid complex	26	160	1.76	<0.001
**KEGG**					
ecb05140	Leishmaniasis	4	104	1.13	0.009
ecb05414	Dilated cardiomyopathy	4	113	1.1	0.011
**Subcellular localization (COMPARTMENTS)**					
GOCC:0005577	Fibrinogen complex	4	11	2.11	<0.001
GOCC:0072562	Blood microparticle	4	13	2.04	<0.001
GOCC:0031091	Platelet alpha granule	3	32	1.52	0.009

**Table 6 animals-15-01604-t006:** Results of random forest model for classification of horses with colic (*n* = 20) by lesion type (ischemic vs. non-ischemic) based on features of the combined peritoneal fluid proteome and clinical data set. Features were selected by the Boruta algorithm: those highlighted in green were classified as “confirmed important” and features in blue were classified as “tentative”. The mean decrease in accuracy (MDA) is a measure of variable importance, reflecting the number of observations that are incorrectly classified by removing the feature.

UniProtKB AC/ID	Feature/Protein	Gene Name	MDA
P02062	Hemoglobin subunit beta	HBB	7.74
P01958	Hemoglobin subunit alpha	HBA2	7.21
A0A3Q2LE47	Inter-alpha-trypsin inhibitor heavy chain 4	ITIH4	6.39
A0A3Q2H333	Serum albumin	ALB	6.61
-	PF color	-	7.22
A0A5F5PYW9	Alpha-1-antiproteinase 2	SPI2	6.23
A0A3Q2HW24	Joining chain of multimeric IgA and IgM	JCHAIN	3.44
F6ZRH8	Uncharacterized protein	ENSECAP00000009652	5.73
A0A5F5PIE1	Attractin	ATRN	2.47
F7CSL8	Alpha-1-antiproteinase 2	SPI2	5.03
F6USP9	Plasmin heavy chain A	PLG	4.61
F6RI47	Alpha-2-macroglobulin	A2M	4.25
-	PF TS	-	3.72
-	L-lactate	-	3.12

**Table 7 animals-15-01604-t007:** Results of random forest model for classification of horses with colic (*n* = 20) by lesion location (large intestine vs. small intestine) based on features of the combined peritoneal fluid proteome and clinical data set. Features were selected by the Boruta algorithm: those highlighted in green were classified as “confirmed important” and features in blue were classified as “tentative”. The mean decrease in accuracy (MDA) is a measure of variable importance, reflecting the number of observations that are incorrectly classified by removing the feature.

UniProtKB AC/ID	Feature/Protein	Gene Name	MDA
A0A3Q2H875	Hemoglobin subunit theta-1	HBA	9.99
P48770	Complement component C9	C9	6.9
A0A3Q2HRX3	Apolipoprotein B	APOB	10.59
	Age		11.82
P14452	Fibrinogen alpha chain	FGA	6.66
A0A5F5PJQ9	Fibrinogen beta chain	FGB	2.68
A0A3Q2H333	Serum albumin	ALB	6.77

## Data Availability

The proteomics data, including raw files, MaxQuant parameters, and results, have been deposited in the ProteomeXchange Consortium via the jPOST partner repository^31^ with the data set identifiers JPST003601 and PXD060655.

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
