# Peer review of "Alterations in the Peritoneal Fluid Proteome of Horses with Colic Attributed to Ischemic and Non-Ischemic Intestinal Disease"

_animals, 2025, doi:10.3390/ani15111604_

Round 1

Reviewer 1 Report

Comments and Suggestions for Authors

General Comments

The challenge of formulating an accurate prognosis in intestinal diseases presenting as colic syndrome in horses remains a constant concern for the attending veterinarian. In this context, it is vital to use humoral markers whose sensitivity and specificity allow for an accurate assessment of the severity of intestinal lesions. Currently, the formulation of a vital prognosis relies on standardized protocols for evaluating horses with colic. These include systematic clinical examination, laboratory tests of blood and peritoneal fluid (e.g., complete blood count, glucose, lactate, fibrinogen, D-dimers), and, when available, imaging techniques (e.g., ultrasonography and radiographic examination). However, their specificity in determining the nature and severity of lesions remains modest, making high-precision prognostic evaluation difficult in many cases.

Given these considerations, the paper titled “Alterations in the peritoneal fluid proteome of horses with intestinal disease” is a valuable study. Although it may be classified as fundamental research, its findings clearly exhibit practical implications with impact on the diagnosis and prognosis of intestinal diseases that induce colic in horses.

The study follows a classical design, including a thorough literature review with 93 relevant sources and a section dedicated to the original research. The study successfully integrates, in a logical sequence, both classical methods (e.g., clinical examination, stall-side blood work, stall-side peritoneal fluid analysis) and advanced laboratory techniques targeting the evaluation of the peritoneal fluid proteome. The criteria for including biological material were rigorous, excluding pathological states (e.g., colitis, primary peritonitis) that could clinically and humoral confound the results. The proteomic analysis of peritoneal fluid presented in this work complements the few studies in the field and successfully identifies 236 proteins in horses with ischemic and non-ischemic intestinal lesions. Some proteins were identified for the first time (e.g., calprotectin, lactotransferrin, alpha-2-macroglobulin, apolipoprotein B). These proteomic alterations were analyzed using advanced statistical methods, lending greater credibility to the findings. Although the authors acknowledge that none of the identified biomarkers can individually predict the clinical trajectory of colic syndrome in horses, they suggest that the results are promising and warrant further studies with larger cohorts.

Specific Comments

  • Lines 2–3: Although the abstract clarifies the pathological states in which peritoneal fluid proteome alterations occur, the title remains too general, creating ambiguity regarding the specific intestinal diseases involved. The term “intestinal disease” may imply a wide range of conditions, including infiltrative, toxic (e.g., NSAID-induced, IgE-mediated, inflammatory or neoplastic diseases. As correctly stated in lines 81–84, it is essential to characterize the proteomic profile in the specific context of intestinal diseases. Therefore, I recommend using terminology that more precisely defines the conditions investigated in this study. As a suggestion, replacing “intestinal disease” with “ischemic or non-ischemic intestinal disease” may be more appropriate. However, given the authors’ deeper understanding of their study’s particularities, I encourage them to find the optimal formulation.
  • Lines 111–112: Besides the listed parameters from the physical examination, did you employ any other diagnostic methods? If so, please specify.
  • Line 117: What method was used to identify the lesion that necessitated laparotomy? Please specify the technique applied.
  • Lines 119–121: It is not sufficiently clear what diagnostic criteria or methods were used to assign horses to the non-ischemic and ischemic groups (S.I., L.I.). In addition to the previously described methods (lines 109–115), did you use other diagnostic techniques? If so, please specify them.
  • Lines 200–201: The sentence “Principal component analysis (PCA) was performed using the ‘MSnSet.utils’ package version 0.2.0.39” should be moved in place of the same phrase in lines 197-198.
  • Line 208: The abbreviation “RF” stands for “Random Forest.” Although it is used as such in the text, its definition is missing from the “Abbreviations” section. I recommend adding it.
  • Lines 248–252 (Table 2): Given the precise diagnoses listed in the table, the term “enteritis” used for horses 14 and 16 is too general. If available, please provide additional details regarding the location, lesion profile, and etiology of the enteritis cases.

Author Response

General Comments

The challenge of formulating an accurate prognosis in intestinal diseases presenting as colic syndrome in horses remains a constant concern for the attending veterinarian. In this context, it is vital to use humoral markers whose sensitivity and specificity allow for an accurate assessment of the severity of intestinal lesions. Currently, the formulation of a vital prognosis relies on standardized protocols for evaluating horses with colic. These include systematic clinical examination, laboratory tests of blood and peritoneal fluid (e.g., complete blood count, glucose, lactate, fibrinogen, D-dimers), and, when available, imaging techniques (e.g., ultrasonography and radiographic examination). However, their specificity in determining the nature and severity of lesions remains modest, making high-precision prognostic evaluation difficult in many cases.

Given these considerations, the paper titled “Alterations in the peritoneal fluid proteome of horses with intestinal disease” is a valuable study. Although it may be classified as fundamental research, its findings clearly exhibit practical implications with impact on the diagnosis and prognosis of intestinal diseases that induce colic in horses.

The study follows a classical design, including a thorough literature review with 93 relevant sources and a section dedicated to the original research. The study successfully integrates, in a logical sequence, both classical methods (e.g., clinical examination, stall-side blood work, stall-side peritoneal fluid analysis) and advanced laboratory techniques targeting the evaluation of the peritoneal fluid proteome. The criteria for including biological material were rigorous, excluding pathological states (e.g., colitis, primary peritonitis) that could clinically and humoral confound the results. The proteomic analysis of peritoneal fluid presented in this work complements the few studies in the field and successfully identifies 236 proteins in horses with ischemic and non-ischemic intestinal lesions. Some proteins were identified for the first time (e.g., calprotectin, lactotransferrin, alpha-2-macroglobulin, apolipoprotein B). These proteomic alterations were analyzed using advanced statistical methods, lending greater credibility to the findings. Although the authors acknowledge that none of the identified biomarkers can individually predict the clinical trajectory of colic syndrome in horses, they suggest that the results are promising and warrant further studies with larger cohorts.

Author Response: Thank you for your thorough summary and reading of our manuscript.

Specific Comments

Lines 2–3: Although the abstract clarifies the pathological states in which peritoneal fluid proteome alterations occur, the title remains too general, creating ambiguity regarding the specific intestinal diseases involved. The term “intestinal disease” may imply a wide range of conditions, including infiltrative, toxic (e.g., NSAID-induced, IgE-mediated, inflammatory or neoplastic diseases. As correctly stated in lines 81–84, it is essential to characterize the proteomic profile in the specific context of intestinal diseases. Therefore, I recommend using terminology that more precisely defines the conditions investigated in this study. As a suggestion, replacing “intestinal disease” with “ischemic or non-ischemic intestinal disease” may be more appropriate. However, given the authors’ deeper understanding of their study’s particularities, I encourage them to find the optimal formulation.

Author Response: Thank you for this valuable feedback. We have changed the title to “Alterations in the peritoneal fluid proteome of horses with colic attributed to ischemic and non-ischemic intestinal disease”.

Lines 111–112: Besides the listed parameters from the physical examination, did you employ any other diagnostic methods? If so, please specify.

Author Response: Change made as suggested.

Line 117: What method was used to identify the lesion that necessitated laparotomy? Please specify the technique applied.

Author Response: Treatment decisions were made by the attending clinician based on clinical evaluation; this has now been clarified. A complete description of the surgical decision-making algorithm for horses with abdominal pain has been well described elsewhere and is beyond the scope of this manuscript.

Lines 119–121: It is not sufficiently clear what diagnostic criteria or methods were used to assign horses to the non-ischemic and ischemic groups (S.I., L.I.). In addition to the previously described methods (lines 109–115), did you use other diagnostic techniques? If so, please specify them.

Author Response: Other routine clinical diagnostic techniques were used by the attending clinician to evaluate horses presenting for abdominal pain; these are now described. Assignment to the groups was made retrospectively based on the diagnosis recorded in the medical record.

Lines 200–201: The sentence “Principal component analysis (PCA) was performed using the ‘MSnSet.utils’ package version 0.2.0.39” should be moved in place of the same phrase in lines 197-198.

Author Response: Change made as suggested.

Line 208: The abbreviation “RF” stands for “Random Forest.” Although it is used as such in the text, its definition is missing from the “Abbreviations” section. I recommend adding it.

Author Response: Change made as suggested.

Lines 248–252 (Table 2): Given the precise diagnoses listed in the table, the term “enteritis” used for horses 14 and 16 is too general. If available, please provide additional details regarding the location, lesion profile, and etiology of the enteritis cases.

Author Response: Unfortunately, a more precise diagnosis for horses 14 and 16 is not available. By definition, enteritis refers to inflammation of the small intestine, and in the majority of horses with a clinical diagnosis of enteritis, no definitive etiologic diagnosis is made. 

Reviewer 2 Report

Comments and Suggestions for Authors

The reviewer would like to thank the authors for the submission of the manuscript titled: Alterations in the peritoneal fluid proteome of horses with intestinal disease 

Please see my remarks:

Consider adding a table/index at the beginning for all of the abbreviations

Line 55: change high to higher risk of post operative surgical complications.

Line 84-86: Break up the objectives into two different sentences. This will make this paragraph read more smooth 

Line 87-91: these two sentences seem misplaced and should be in the materials and methods not in the introduction unless the methodology that you are using is completely different than the only other known cited paper. If so this needs set in its own paragraph. 

Line 101: how did you decide to exclude colitis and primary peritonitis but not enteritis?

Line 113: change lactate to L-lactate....

Line 113: change to total peripheral solids from total solids

How did you control for lesion variation and severity since you normalized the total enrollment for the study to be 20 horses (5 in each group). Did you just take the first 5 that met each criteria?

Line 120: add presumptive diagnosis or diagnosis after lesion type (ie some medical treatments cannot be 100% ascertained without surgery)

Line 128: describe where the Roy J. Carver.... is etc. 

Line 131: what is the BCA assay.... describe more

Line 154: what is NCE....

Line 167: what is PSM.... be consistent with abbreviations or make a table to state what each of them are at the beginning of the manuscript rather than at the end or make a footnote guiding the reader to the abbreviations section

Line 219: may consider the use of small intestinal strangulating obstructive lesion rather than strangulating lipoma unless all were strangulating lipomas. If so state this. 

Line 221: did you quantify SIRs criteria any other way ie with the horses that had increased CRT was the HR, temp etc elevated. There are categorical criteria for SIRs to be met to diagnose 

The results and discussion are confusing to read based on how they are laid out. Is it possible to break it down into a true results section and then a true discussion section rather than having a mini discussion for each results or consider having a title for results and discussion SI and Results and discussion LI. Within each of these could be strangulating vs non strangulating. 

Line 553: consider classifying this paragraph as limitations to the study. 

Author Response

Consider adding a table/index at the beginning for all of the abbreviations

Author Response: Abbreviations are summarized at the end of document as per journal format guidelines.

Line 55: change high to higher risk of post operative surgical complications.

Author Response: Change made as suggested.

Line 84-86: Break up the objectives into two different sentences. This will make this paragraph read more smooth

Author Response: We appreciate your suggestion but feel that it would be overly repetitive to restructure the objectives into two separate sentences. We will defer to the editor for their preference on this matter.

Line 87-91: these two sentences seem misplaced and should be in the materials and methods not in the introduction unless the methodology that you are using is completely different than the only other known cited paper. If so this needs set in its own paragraph.

Author Response: We feel it is relevant to briefly summarize the methodology in the final paragraph of our introduction as this type of study can be affected by the approach to protein quantitation. However, we will defer to the editor if they wish this information to be moved elsewhere.

Line 101: how did you decide to exclude colitis and primary peritonitis but not enteritis?

Author Response: Colitis and primary peritonitis have distinct presentations that are readily distinguished from other causes of colic with standard clinical evaluation, whereas enteritis often has a similar presentation to other types of small intestinal colic.

Line 113: change lactate to L-lactate....

Author Response: Change made as suggested.

Line 113: change to total peripheral solids from total solids

Author Response: Total solids is the standard name for the clinical parameter that was measured.

How did you control for lesion variation and severity since you normalized the total enrollment for the study to be 20 horses (5 in each group). Did you just take the first 5 that met each criteria?

Author Response: Yes, the first 5 samples in each category that met the criteria were included.

Line 120: add presumptive diagnosis or diagnosis after lesion type (ie some medical treatments cannot be 100% ascertained without surgery)

Author Response: Change made as suggested.

Line 128: describe where the Roy J. Carver.... is etc.

Author Response: The Roy J. Carver Biotechnology Center is located at the University of Illinois; this has now been added.

Line 131: what is the BCA assay.... describe more

Author Response: Change made as suggested.

Line 154: what is NCE....

Author Response: Change made as suggested.

Line 167: what is PSM.... be consistent with abbreviations or make a table to state what each of them are at the beginning of the manuscript rather than at the end or make a footnote guiding the reader to the abbreviations section

Author Response: Change made as suggested.

Line 219: may consider the use of small intestinal strangulating obstructive lesion rather than strangulating lipoma unless all were strangulating lipomas. If so state this.

Author Response: While not all SI strangulations in our study were lipomas (as stated in table 2), we are commenting here on the overall increase in age compared to other groups, and the association in the literature between aged horses and strangulating lipoma specifically.

Line 221: did you quantify SIRs criteria any other way ie with the horses that had increased CRT was the HR, temp etc elevated. There are categorical criteria for SIRs to be met to diagnose

Author Response: No, SIRS scores were not specifically quantified in this study. We are simply commenting on a likely explanation for the difference in CRT. This has been revised for clarity.

The results and discussion are confusing to read based on how they are laid out. Is it possible to break it down into a true results section and then a true discussion section rather than having a mini discussion for each results or consider having a title for results and discussion SI and Results and discussion LI. Within each of these could be strangulating vs non strangulating.

Author Response: Per the journal formatting instructions, results and discussion may be combined into a single section. We will defer to the editor’s discretion if a major reorganization is necessary.

Line 553: consider classifying this paragraph as limitations to the study.

Author Response: Change made as suggested.

Reviewer 3 Report

Comments and Suggestions for Authors

Dear Authors, it is a very intersting manuscript. Based on our results horses withs large colon torsion have much worse prognosis than horses with strangulation obstruction of the small in intestine. You should include this!

-------------------------

Additional comments:

  • Specific proteome in the peritoneal fluid in horses with a specific intestinal lesion is the main question addressed by the research.
  • This is important because the veterinarian can decide in the field the horse has a strangulation obstruction in the small intestine or in a torsion of the large colon.
  • Lactate concentration in the peritoneal fluid was added to the subject area compared with other published material.
  • If the this method could be used in the field, it would be a great progress!
  • The conclusions are consistent with the evidence and arguments presented and they address the main question posed.
  • The references are appropriate.
  • There are no additional comments on the tables and figures.

Author Response

Dear Authors, it is a very interesting manuscript. Based on our results horses withs large colon torsion have much worse prognosis than horses with strangulation obstruction of the small in intestine. You should include this!

Author Response: Thank you for your review and feedback. While we don’t disagree with your experience regarding colon torsions and small intestinal strangulations, the included cases in this study do not support rigorous evaluation of prognosis, as mentioned in the limitations. If there is place within the manuscript where you feel discussion of LCV/SISO prognosis is appropriate, please advise.

-------------------------

Additional comments:

Specific proteome in the peritoneal fluid in horses with a specific intestinal lesion is the main question addressed by the research.

This is important because the veterinarian can decide in the field the horse has a strangulation obstruction in the small intestine or in a torsion of the large colon.

Lactate concentration in the peritoneal fluid was added to the subject area compared with other published material.

If the this method could be used in the field, it would be a great progress!

The conclusions are consistent with the evidence and arguments presented and they address the main question posed.

The references are appropriate.

There are no additional comments on the tables and figures.

Round 2

Reviewer 2 Report

Comments and Suggestions for Authors

The authors have done an excellent job presenting their research and in the present form it is ready for publication.